# Innovative qPCR Algorithm Using Platelet-Derived RNA for High-Specificity and Cost-Effective Ovarian Cancer Detection

**DOI:** 10.3390/cancers17071251

**Published:** 2025-04-07

**Authors:** Eunyong Ahn, Se Ik Kim, Sungmin Park, Sarah Kim, Hyunjung Kim, Hyejin Lee, Heeyeon Kim, Eun Ji Song, TaeJin Ahn, Yong-Sang Song

**Affiliations:** 1Foretell My Health, Inc., 558 Handong-ro Buk-gu, Pohang 37554, Republic of Korea; 2Department of Obstetrics and Gynecology, Seoul National University College of Medicine, Seoul 03080, Republic of Korea; 3Cancer Research Institute, Seoul National University College of Medicine, Seoul 03080, Republic of Korea; 4Department of Advanced Convergence, Handong Global University, Pohang 37554, Republic of Korea; 5Department of Obstetrics and Gynecology, Myongji Hospital, Goyang 10475, Republic of Korea

**Keywords:** ovarian cancer, platelet RNA, qPCR algorithm, early detection, inflammation

## Abstract

Ovarian cancer is one of the deadliest gynecologic cancers, largely due to the difficulty of detecting it early. Current screening methods, such as CA125 blood tests and ultrasound, lack accuracy, while advanced genetic tests are costly and impractical for widespread use. This study aims to develop a cost-effective and accessible diagnostic method using qPCR-based platelet RNA profiling to detect ovarian cancer earlier, particularly the aggressive high-grade serous ovarian cancer (HGSOC). By analyzing blood samples, we identified RNA-based biomarkers that can differentiate ovarian cancer from benign conditions with over 94% accuracy. These findings suggest that platelet RNA biomarkers could improve early detection, potentially leading to better survival rates. This research contributes to the advancement of liquid biopsy diagnostics, offering a promising alternative to current screening approaches.

## 1. Introduction

Ovarian cancer remains one of the most lethal gynecologic malignancies, with high-grade serous ovarian cancer (HGSOC) being its most aggressive and prevalent subtype. Early detection of localized ovarian cancer could ensure successful treatment for over 90% of affected women. However, once the cancer metastasizes, the 5-year survival rate drops dramatically to less than 30%. Current statistics reveal that approximately 75% of ovarian cancer cases are diagnosed at advanced stages (III or IV), primarily due to the absence of disease-specific symptoms and effective screening tools [1].

Current ovarian cancer screening tests are limited by suboptimal sensitivity and specificity. The UK Collaborative Trial of Ovarian Cancer Screening (UKCTOCS), which enrolled 202,638 postmenopausal women between 2001 and 2005, investigated whether population screening could reduce mortality rates. While multi-modal screening (MMS) using longitudinal CA125 and transvaginal ultrasound led to a modest reduction in late-stage (III/IV) disease incidence, this reduction did not translate into reduced mortality, indicating that general population screening may not be clinically beneficial [2,3].

Recent clinical trials have continued to explore alternative screening approaches. The Normal Risk Ovarian Screening Study (NROSS) investigated a two-stage screening strategy that first applied the Risk of Ovarian Cancer Algorithm (ROCA)—a computer-based tool that calculates an individual’s risk level based on CA125 trends. In the second stage, patients classified as high-risk underwent transvaginal ultrasound (TVS) for further evaluation. This personalized approach significantly increased early-stage ovarian cancer detection, achieving a high positive predictive value (50%) and leading to a 30–34% reduction in late-stage diagnoses compared to other cohorts. These findings underscore the potential of biomarker-driven screening in enhancing early ovarian cancer detection [4]. Additionally, the FORECEE (4C) consortium has explored the use of epigenetic markers in cervical samples for ovarian cancer risk prediction, though validation in larger populations is still needed [5].

Existing biomarker-driven approaches have primarily focused on genetic and protein markers; however, growing evidence suggests that inflammation-driven molecular changes also play a crucial role in ovarian cancer progression. The concept of “inflamm-aging”, introduced by Franceschi in 2000, describes immune changes in response to lifelong stress. This chronic low-grade inflammation associated with aging plays a key role in ovarian cancer development by fostering a tumor-promoting environment [6,7,8]. In ovarian tissue, inflamm-aging manifests through repeated trauma to the ovarian epithelium during ovulation, alongside age-related hormonal changes that alter the inflammatory environment. This process is compounded by accumulated DNA damage, impaired repair mechanisms, and compromised immune surveillance systems [6,9,10].

Given their active role in immune modulation and inflammation, platelets are increasingly recognized as key mediators in the cancer-inflammation relationship axis [11]. These cells engage in complex molecular interactions with tumor cells, where tumor-educated platelets undergo specific RNA splicing events in response to cancer signals [12,13]. The interaction involves direct transfer of tumor-derived RNA through microvesicles, while platelets regulate tumor blood vessel formation by releasing specific factors. These cancer-specific changes in platelet RNA profiles effectively mirror the body’s systemic response to malignancy, providing valuable diagnostic information.

Recent advances in platelet RNA analysis have demonstrated remarkable potential for early tumor detection. Studies have shown that platelet RNA sequencing signatures can detect ovarian cancer with the area under the curve (AUC) 0.918, indicating strong diagnostic potential. Combining tumor-educated platelets (TEP) analysis with the CA125 biomarker further improved detection accuracy [14]. Notable alterations in platelet count, structure, and inflammatory markers have been observed in patients with epithelial ovarian cancer, suggesting their utility as diagnostic tools [15,16].

Our study addresses the challenges of early ovarian cancer detection by developing a cost-effective, qPCR-based diagnostic approach that utilizes platelet RNA profiling for accurate cancer detection. We diverge from traditional methods by employing intron-spanning read (ISR) counts rather than conventional gene expression levels. This innovative approach enhances detection of cancer-specific splicing events while reducing interference from contaminating genomic DNA. The method significantly improves quantification of low-abundance transcripts and enables better discrimination of tumor-specific RNA signatures. Most importantly, it provides higher sensitivity for detecting subtle molecular changes associated with early-stage disease, making it a powerful tool for early cancer diagnosis.

The ISR approach allows for more precise detection of splice junction variations associated with cancer development, potentially enabling earlier and more accurate diagnosis than conventional methods. This technique specifically captures alternative splicing events that may serve as early indicators of malignant transformation, providing a new dimension to molecular diagnostics in ovarian cancer.

## 2. Materials and Methods

### 2.1. Patient Recruitment and Blood Sample Collection

Peripheral blood samples were prospectively collected from patients enrolled in this study between August 2022 and January 2025 at Seoul National University Hospital (SNUH), Myongji Hospital (MJH), and Boaz Medical Center at Handong Global University (HGU). To minimize potential confounding factors that could impact the study outcomes, specific exclusion criteria were applied as follows: patients with cancers originating from non-gynecological tissues were excluded from the study. Patients who were currently taking or had taken (within the past 7 days) hormonal therapy, anti-coagulants, or non-steroidal anti-inflammatory drugs (NSAIDs) were excluded. Patients who had received chemotherapy or radiation therapy within the past 7 days were excluded to prevent potential treatment-related bias. Patients who had previously undergone surgical resection of gynecological tumors were excluded, as their disease status was not aligned with the study’s objective. For the asymptomatic control group, individuals with suspicious symptoms indicative of an infectious disease were excluded to maintain the integrity of the control cohort. After applying these exclusion criteria, the final study cohort comprised 17 ovarian cancer patients, 2 borderline ovarian tumor patients, 37 patients with benign tumors, and 34 asymptomatic female controls. The metadata of the dataset is summarized in Appendix A.

### 2.2. Platelet RNA Extraction

Blood samples were collected using 10 mL EDTA-coated purple-capped BD Vacutainers (BD) and stored at 4 °C until further processing. Platelets were isolated within 48 h post-collection using a two-step centrifugation process, according to the previously established protocol to ensure consistency across samples [13]. The extracted platelets were suspended in RNAlater (Thermo Fisher Scientific, Waltham, MA, USA) and stored at 4 °C overnight, followed by long-term storage at −80 °C in a deep freezer. Total RNA was extracted within two months using the mirVana RNA Isolation Kit (Thermo Fisher Scientific, Waltham, MA, USA).

### 2.3. RNA Sequencing

Total RNA quality was assessed using BioAnalyzer 2100 (Agilent Technologies, Santa Clara, CA, USA), and RNA samples with an RNA Integrity Number (RIN) ≥ 6 or a distinct ribosomal peak were considered of high quality for sequencing. The purity of RNA samples is described in Appendix A. For RNA sequencing, 500 pg of platelet RNA was subjected to cDNA synthesis and amplification using the SMART-Seq v4 Ultra Low Input RNA Kit (Takara Bio Inc., San Jose, CA, USA), with quality control performed using BioAnalyzer 2100. The amplified cDNA underwent fragmentation via Covaris ultrasonication, followed by the attachment of Illumina sequencing index barcodes using the Truseq Nano DNA Sample Prep Kit platform (Illumina, Inc., San Diego, CA, USA). Subsequent purification was performed using AMPure XP beads (Beckman Coulter Life Sciences, Indianapolis, IN, USA) after eight cycles of PCR amplification. The final library concentration and fragment size distribution were evaluated using TapeStation 4200 (Agilent, Santa Clara, CA, USA). Libraries were pooled within the 500–600 bp size range and sequenced on the Illumina NovaSeq6000 platform (Illumina, Inc., San Diego, CA, USA) using 150 bp paired-end sequencing.

### 2.4. Data Processing and Quantification

To improve the quality of RNA sequencing data and ensure accurate expression analysis, multiple preprocessing steps were performed. Initially, Trimmomatic (v.0.39) was used for adapter removal and quality-based read trimming. Following this, HISAT2 (v.2.1.0) was employed to align the filtered sequencing reads to the human GRCh38 reference genome. The resulting SAM files were converted to BAM format, and only primary alignments were retained using Samtools (v.1.9). The BAM files were then utilized to calculate gene-level expression in FPKM and TPM, as well as junction-level expression in CPM, based on the following criteria: reads spanning 150 bp upstream and 150 bp downstream of a given splice junction; reads containing a CIGAR string with an N, indicating a splicing event; reads with spliced regions aligning precisely with the annotated junction site. This process yielded FPKM and TPM values for 60,624 genes and CPM values for 2,855,955 splice junctions in the clinical RNA-seq dataset. FPKM and TPM calculations were performed using StringTie (v.2.1.7).

### 2.5. Quantitative Real-Time Polymerase Chain Reaction

For RT-PCR analysis, cDNA was synthesized from platelet RNA using the PrimeScript™ RT reagent kit with gDNA (Takara Bio Inc., San Jose, CA, USA). Genomic DNA was removed by incubating with gDNA eraser at 42 °C for 2 min, followed by reverse transcription at 37 °C for 15 min and inactivation at 85 °C for 5 s.

For qPCR analysis, the cDNA was diluted 1:100 and used as a template. Reactions were performed with the FastFACT™ 2X qPCR master mix (BIOFACT Co., Ltd., Daejeon, Republic of Korea) in a total volume of 20 µL, including 0.6 µL each of primers and probe, and 4.2 µL RNase-/DNase-free water. qPCR was conducted using the CFX Opus 96 Real-Time PCR System (Bio-Rad, Hercules, CA, USA). The thermal cycling conditions included an initial denaturation at 50 °C for 2 min and 95 °C for 10 min, followed by a pre-cycling phase (5 cycles at 95 °C for 20 s and 65 °C for 1 min) and amplification (40 cycles at 95 °C for 20 s and 60 °C for 1 min).

### 2.6. Junction-Level RNA Sequencing Analysis and Marker Selection

Aberrant RNA splicing in cancer cells can generate unique splice junctions, serving as potential biomarkers. This study aimed to identify ovarian cancer-specific junction markers using junction-level RNA sequencing data. The dataset used for marker selection comprised of 32 asymptomatic controls, 16 patients with benign ovarian tumors, and 13 ovarian cancer patients. To normalize the sequencing data, junction-level CPM values were log-transformed (log_2_CPM) before analysis. Candidate markers were identified based on the following criteria: At least 70–95% of asymptomatic and benign samples exhibited log_2_CPM values below predefined thresholds ([0.2, 0.3, 0.4, 0.5]). At least one ovarian cancer sample exhibited significantly high expression of the junction. Final candidate markers were validated using PCR experiments, and the correlation between sequencing and PCR data was assessed. Markers with no expression level in asymptomatic control and benign groups or an R-squared value of 0.4 or higher were selected, resulting in 10 validated markers.

### 2.7. Cutoff and Score for the Algorithm

A classification algorithm was developed and validated using PCR data to differentiate ovarian cancer from non-cancerous cases. The training dataset used for marker selection, excluding asymptomatic controls, was employed for PCR analysis and algorithm development, while an independent test dataset was used for validation. The training dataset consisted of 16 patients with benign tumors and 13 ovarian cancer patients, while the test dataset comprised 21 benign tumor patients, 4 ovarian cancer patients, and 34 asymptomatic controls. Although asymptomatic controls, except 2 samples, were utilized for marker selection, they were not included in model development and were instead used exclusively for validation. Additionally, two patients with borderline ovarian tumors were included for validation.

Each marker was measured twice per sample. If the qPCR Cq value was undetectable, it was assigned a value of 41, and the average of the duplicate measurements was used. Non-specific signal cutoffs were determined for each marker using benign tumor samples from the training dataset, and Cq values exceeding these cutoffs were replaced with 41. Final qPCR Cq values were normalized using the ΔCq method (Cq_target—Cq_ACTB).

Scoring cutoffs were established for seven individual markers and one composite variable (sum of five markers) using the training dataset to achieve 99% specificity. Cutoff values were determined based on the mean ΔCq value at the benign tumor–ovarian cancer threshold. Scores were subsequently calculated by assessing the differences between observed PCR values and the marker-specific cutoffs.

For each marker *m* in sample *s*, the difference from its cutoff value was calculated using the following formula:xs,m=cutoffm−observed adjusted Cq value from PCR of s

For the marker combination IL1R2 and DEFA, the formula was modified as follows:xs,m=(cutoffm−observed adjusted Cq value from PCR of s)/5

The calculated values were transformed using the following function:Fxs,m=tanh⁡xs,m if xs,m≥0xs,m10              if xs,m<0

Each marker was assigned a weight based on its Area Under the Curve (AUC), determined using the training dataset:wm=AUCm

For samples where at least one *F**x**s*, > 0, any *F**x**s*, < 0 values were replaced with 0.

For each sample *s*, only non-zero *F**x**s*,*m* values were selected. The final score was calculated by applying weights to the selected *F**x**s*,*m* values and averaging the results:Total Scores=∑m∈Ms(Fxs,m×wm)|Ms|Ms=mFxs,m≠0

The final classification was made based on the Total Score, where scores ≥ 0 were classified as ovarian cancer, and scores ≤ 0 were classified as non-ovarian cancer.

### 2.8. Quantification and Statistical Analysis

Model performance was evaluated using accuracy, sensitivity, specificity, balanced accuracy, and AUC of the ROC curve. Statistical computations were performed using scikit-learn (v.1.1.1) in Python. To assess the reliability of AUC values, 95% confidence intervals were calculated using Wilson’s method. Statistical analysis and data visualization were conducted using NumPy, pandas, and matplotlib in Python (v. 3.8.13).

## 3. Results

### 3.1. Study Population and Classification

A total of 90 women from three different medical institutions were enrolled in the study between August 2022 and January 2025, and were categorized into asymptomatic control, gynecological benign tumor, and ovarian cancer (OC) groups. The OC group was further classified by stage (I–IV) and included borderline ovarian tumors (BOTs). BOTs represent a heterogeneous group of neoplasms with recognized potential malignancy. While the prognosis is generally excellent, a subset of patients may develop a more aggressive course, ultimately leading to mortality. Given this malignant potential, we have categorized borderline ovarian tumors alongside ovarian cancer in our analysis. Asymptomatic control (AC) samples were primarily obtained from Handong Global University Boaz Medical Center. This resulted in a younger median age in the Asymptomatic control group (25.5 years, IQR: 22.2–33.0), compared to 46.0 years (IQR: 38.2–52.0) in Benign and 53.5 years (IQR: 50.5–61.8) in OC (*p* = 0.0001). Weight varied across groups, with medians of 56.0 kg (Asymptomatic control), 65.8 kg (Benign), and 59.2 kg (OC) (*p* = 0.691). Height ranged from 159.8 cm (OC) to 162.1 cm (Benign), with no significant differences between Asymptomatic control, Benign, and OC groups (*p* = 0.104). BMI was slightly higher in the Benign group (25.3) compared to Asymptomatic controls (21.7) and OC (23.6) but was not statistically significant (*p* = 0.208). While statistical comparisons are reported, *p*-values in the demographic profile are provided for descriptive purposes rather than inferential conclusions, as the primary aim is to characterize the study population rather than test hypotheses on demographic variables. The clinical characteristics of study participants are summarized in Table 1.

### 3.2. Assessment of Batch Effects

To assess potential batch effects resulting from sample collection at multiple institutions, we examined the expression levels of ACTB, a reference marker, in both sequencing and qPCR data. In the sequencing data, the log2CPM values of ACTB across 61 training samples ranged from 12.70 to 14.05, showing no significant differences (Figure 1A). Similarly, in the qPCR data, the ACTB Cq values across all samples ranged from 17.25 to 20.88, with no significant differences observed among the three institutions (Figure 1B). The narrow and consistent distribution of ACTB Cq values further supports reliable qPCR efficiency.

### 3.3. Dataset Composition and Marker Selection

Among the 90 total samples, 61 samples (13 OC, 16 Benign, and 32 AC) were included in the training dataset for marker selection. Only the OC and Benign group from the training dataset were used for model development. The total 34 samples in AC group, including two samples that were not used for marker selection, were utilized for validation. An independent set of 27 samples (four OC, 21 Benign, and two BOT) was used as the test dataset for model evaluation. We analyzed sequencing data of the training dataset to identify candidate markers that were absent in the control group (benign tumors and asymptomatic controls) but exhibited outlier expression in some ovarian cancer samples. With the exception of IL1R2_a (we noted a list of markers that follow the gene name with the suffix “_a” or “_b”), DEFA1_a, and CD177_a, the selected markers did not show significant differences between the ovarian cancer and benign tumor groups. However, at least two ovarian cancer samples displayed distinctly higher expression levels for each marker compared to most control samples. Notably, DEFA1B_a and CD177_a were completely absent in all control group samples but were exclusively expressed in a subset of ovarian cancer cases (Figure 2).

### 3.4. qPCR Validation of Candidate Markers

To validate whether the candidate markers identified through sequencing showed consistent expression patterns in qPCR, we performed linear regression analysis between sequencing data (scaled log2CPM) and qPCR data (reversed ΔCq) to assess correlation (R^2^ values). Among the 10 final markers, some were absent in the control group, while the correlation between sequencing and qPCR data ranged from 0.44 to 0.98 (Figure 3). Although ΔCq values did not show statistically significant differences between ovarian cancer and benign tumor groups, some ovarian cancer samples exhibited lower ΔCq values than benign tumors, aligning with sequencing results (Figure 4). This suggests that these markers are not artifacts of sequencing noise but are genuinely and specifically expressed in ovarian cancer.

### 3.5. Algorithm Development and Performance Evaluation

We developed an algorithm to distinguish ovarian cancer from benign tumors and asymptomatic controls using the ΔCq values of the selected markers. In the training dataset (*n* = 29), the algorithm achieved a sensitivity of 92.3%, specificity of 100.0%, and an AUC of 0.938 (95% CI: 0.790–0.984) (Table 2, Figure 5A and Figure 6). Additionally, among 34 asymptomatic controls who were included in the marker selection process but not in algorithm development, 33 were correctly classified as controls, yielding a specificity of 97.1%.

In the test dataset (*n* = 25), which was not used for either marker selection or algorithm development, the algorithm demonstrated a sensitivity of 100.0%, specificity of 85.7%, and an AUC of 0.887 (95% CI: 0.746–0.976), accurately identifying all ovarian cancer samples (Table 3, Figure 5B and Figure 6). Additionally, one of the two BOT samples was classified under the ovarian cancer group. The overall performance across the training and test datasets yielded a sensitivity of 94.1%, specificity of 94.4%, and an AUC of 0.933 (95% CI: 0.861–0.969) (Table 4, Figure 5C and Figure 6). By implementing pre-defined cutoff values for each marker and a score calculation method, we achieved high sensitivity and specificity.

## 4. Discussion

This study presents a novel, highly specific qPCR-based diagnostic algorithm for ovarian cancer detection, addressing the limitations of current screening methods. The selected biomarkers represent key biological pathways involved in ovarian cancer development and progression. Notably, the inclusion of inflammatory mediators, such as IL1R2, reflects the critical role of inflammation in ovarian cancer pathogenesis, particularly in the context of tumor–platelet interactions. The defensin family genes (DEFA1, DEFA1B, DEFA3) and CD177, typically linked to neutrophil function, may indicate the involvement of innate immunity in cancer development [17,18]. IL1R2, a decoy receptor that inhibits IL-1 signaling, has been implicated in immune suppression within the tumor microenvironment [19,20]. Its upregulation may contribute to cancer immune evasion by dampening inflammatory responses and reducing anti-tumor immunity, a phenomenon commonly observed in ovarian cancer. Similarly, FMO2, previously identified as a significant protein biomarker for ovarian cancer, may contribute to metabolic reprogramming, as recent studies suggest its involvement in oxidative stress regulation and xenobiotic metabolism [21]. The selection of NPRL3, a component of the GATOR1 complex involved in mTOR signaling regulation, aligns with known dysregulation of growth signaling pathways in cancer [22]. Additionally, TCN2, crucial for vitamin B12 transport, may indicate altered metabolic requirements of cancer cells [23].

A major advantage of this approach is its cost-effectiveness compared to NGS-based platforms, making it more accessible for widespread clinical use. The method also demonstrated high specificity (94.4%) and sensitivity (94.1%), outperforming conventional CA125-based screening while maintaining robust performance across samples of varying RNA quality, which is crucial for real-world application. Additionally, its streamlined workflow enhances its feasibility for clinical adoption as a minimally invasive diagnostic tool.

Despite these advantages, this study has several limitations. The number of ovarian cancer patients in the test set was small (*n* = 4), reflecting both the exploratory nature of this study and the challenges in acquiring well-characterized platelet RNA samples from early-stage patients. This limited sample size constrains the statistical power of the findings and emphasizes the need for further validation in larger, independent cohorts. In addition, the control group consisted of asymptomatic individuals who had not undergone systematic ovarian neoplasm screening, raising the risk of misclassification bias due to potential undiagnosed benign or malignant conditions. Another important consideration is the age difference between the cancer and control groups, which was not accounted for in the initial study design. While we conducted a secondary analysis incorporating age as a covariate and observed consistent classifier performance, the lack of age matching remains a confounding factor that should be addressed in future work. Furthermore, the test set was not fully independent, as some control samples used in the marker discovery phase were also included in the classification evaluation. This partial overlap may lead to an overestimation of diagnostic performance, underscoring the importance of validating the algorithm on fully independent datasets. With regard to the inclusion of borderline ovarian tumors, two patients were included in the validation set. While only one of these cases was correctly classified, their inclusion reflects the clinical relevance of borderline tumors and their importance in evaluating diagnostic performance. Technical challenges inherent to platelet RNA profiling, such as platelet activation, RNA degradation, and contamination with leukocyte RNA, also pose barriers to clinical implementation. Addressing these issues will require standardized protocols for sample collection, RNA extraction, and data normalization to ensure reproducibility. Lastly, although our algorithm demonstrated strong diagnostic potential, its broader applicability remains to be tested. Multi-center validation across diverse populations is needed to confirm its robustness, and the potential influence of comorbid conditions and systemic inflammation on platelet RNA expression warrants further investigation to refine biomarker selection and clinical utility.

## 5. Conclusions

Our findings demonstrate the successful development of a highly specific PCR-based diagnostic algorithm for ovarian cancer detection. The platform shows promise for HGSOC detection through platelet RNA profiling, offering a cost-effective alternative to NGS-based approaches. Future directions include expanding sample diversity and exploring integration with complementary biomarker panels.

## Figures and Tables

**Figure 1 cancers-17-01251-f001:**
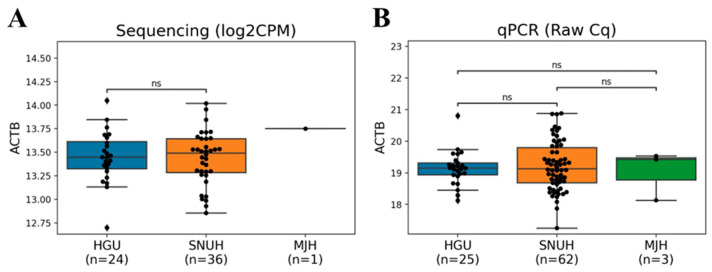
Boxplots of the expression levels of ACTB for institutions. The log_2_CPM and Cq values of samples from Handong Global University Boaz Medical Center (HGU), Seoul National University Hospital (SNUH), Myongji Hospital (MJH) are shown. *p*-values from Wilcoxon tests: ns = not significant. (**A**) The log_2_CPM values of ACTB in training dataset (**B**) The Cq values of ACTB in a total of 90 samples.

**Figure 2 cancers-17-01251-f002:**
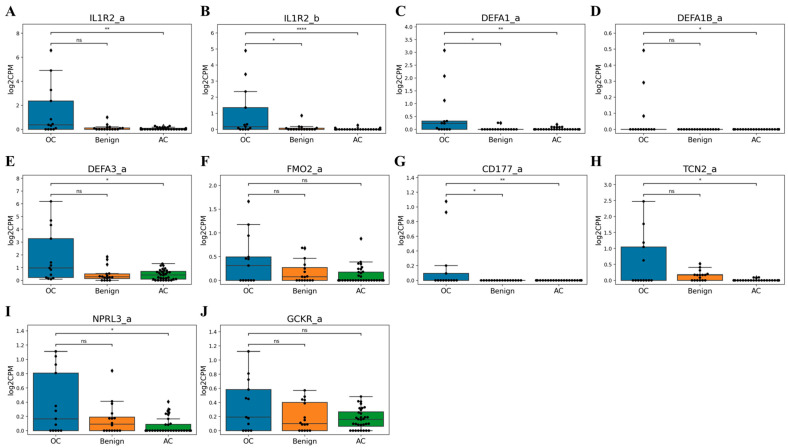
Marker analysis of log2CPM values in the training dataset. Log2CPM values of RNA sequencing data from Asymptomatic control (AC), Benign gynecological tumor (Benign), and Ovarian cancer (OC) group in training dataset are shown. *p*-values from Wilcoxon tests: ns = not significant, * ≤ 0.05, ** ≤ 0.01, **** ≤ 0.0001. (**A**) The expression levels of IL1R2_a marker (**B**) The expression levels of IL1R2_b marker (**C**) The expression levels of DEFA1_a marker (**D**) The expression levels of DEFA1B_a marker (**E**) The expression levels of DEFA3_a marker (**F**) The expression levels of FMO2_a marker (**G**) The expression levels of CD177_a marker (**H**) The expression levels of TCN2_a marker (**I**) The expression levels of NPRL3_a marker (**J**) The expression levels of GCKR_a marker.

**Figure 3 cancers-17-01251-f003:**
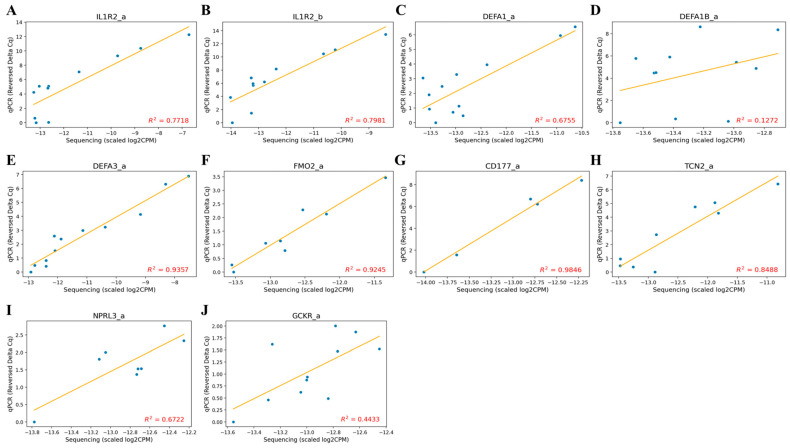
Marker correlation plot of scaled log2CPM values and reversed ΔCq values. Correlation plots of RNA sequencing data and qPCR data are shown. The orange line represents the linear regression line, and the R-squared value is displayed for each marker. Scaled log2CPM values are calculated as the marker log2CPM values minus the ACTB log2CPM values. Reversed ΔCq values are derived by subtracting the marker ΔCq values from the maximum ΔCq value of the marker. Only samples with non-NA raw Cq values are included. (**A**) The correlation plot of IL1R2_a marker (**B**) The correlation plot of IL1R2_b marker (**C**) The correlation plot of DEFA1_a marker (**D**) The correlation plot of DEFA1B_a marker (**E**) The correlation plot of DEFA3_a marker (**F**) The correlation plot of FMO2_a marker (**G**) The correlation plot of CD177_a marker (**H**) The correlation plot of TCN2_a marker (**I**) The correlation plot of NPRL3_a marker (**J**) The correlation plot of GCKR_a marker.

**Figure 4 cancers-17-01251-f004:**
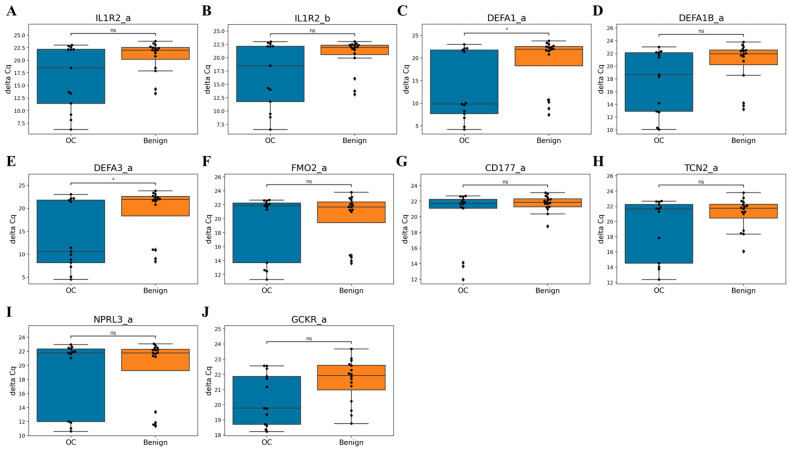
Marker analysis of ΔCq values in training dataset. ΔCq values of qPCR data from Benign gynecological tumor (Benign), and Ovarian cancer (OC) group in training dataset are shown. *p*-values from Wilcoxon tests: ns = not significant, * ≤ 0.05. (**A**) The expression levels of IL1R2_a marker (**B**) The expression levels of IL1R2_b marker (**C**) The expression levels of DEFA1_a marker (**D**) The expression levels of DEFA1B_a marker (**E**) The expression levels of DEFA3_a marker (**F**) The expression levels of FMO2_a marker (**G**) The expression levels of CD177_a marker (**H**) The expression levels of TCN2_a marker (**I**) The expression levels of NPRL3_a marker (**J**) The expression levels of GCKR_a marker.

**Figure 5 cancers-17-01251-f005:**
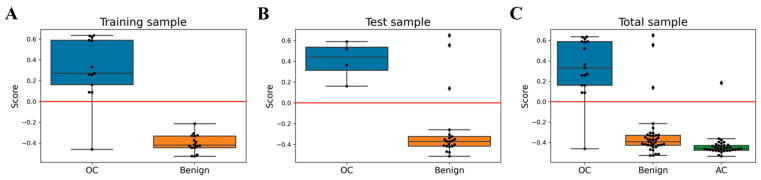
Boxplots of the classification score in training, test, and total dataset. Separate boxplots show the classification scores generated by the algorithm for (**A**) training, (**B**) test, and (**C**) total samples. The groups—Asymptomatic Controls (AC), Benign Gynecological Tumors (Benign), and Ovarian Cancer (OC)—are indicated. The score threshold is represented by a red line.

**Figure 6 cancers-17-01251-f006:**
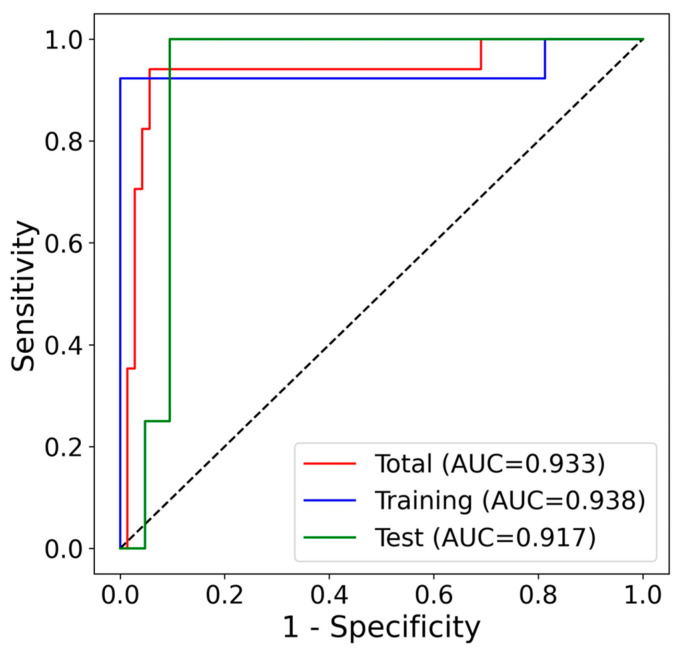
Receiver operating characteristic (ROC) curves of the algorithm.

**Table 1 cancers-17-01251-t001:** Clinical and Demographic Characteristics of Study Participants.

Group	Stage	*n*	Age	Weight	Height	BMI
Asymptomatic control	N.A. *	34	25.5 [22.2, 33.0], *n* = 34	56.0 [52.3, 60.0], *n* = 34	160.6 [159.5, 163.0], *n* = 34	21.7 [20.5, 22.6], *n* = 34
Benign	N.A.	37	46.0 [38.2, 52.0], *n* = 36	65.8 [59.4, 73.5], *n* = 16	162.1 [159.0, 163.8], *n* = 16	25.3 [22.5, 27.8], *n* = 16
Ovarian cancer (+Borderline ovarian tumors)	Total	19	53.5 [50.5, 61.8], *n* = 18	59.2 [54.9, 67.6], *n* = 14	159.8 [154.9, 160.9], *n* = 14	23.6 [22.2, 26.4], *n* = 14
Ovarian cancer	Early (I, II)	6	52.5 [50.5, 53.0], *n* = 6	68.2 [61.1, 69.2], *n* = 5	160.2 [160.1, 161.5], *n* = 5	26.6 [23.4, 27.0], *n* = 5
	Late (III, IV)	11	61.0 [52.5, 61.8], *n* = 10	55.3 [52.6, 62.8], *n* = 8	156.3 [154.5, 160.7], *n* = 8	22.6 [21.4, 24.3], *n* = 8
Borderline ovarian tumors	N.A.	2	53.0 [43.0, 63.0], *n* = 2	57.2 [57.2, 57.2], *n* = 1	149.0 [149.0, 149.0], *n* = 1	25.8 [25.8, 25.8], *n* = 1
Asymptomatic control and Benign vs. Ovarian cancer (+Borderline ovarian tumors) (*p*-value)	-	-	0.0001	0.690588	0.103928	0.208145

Values represent the median, with the interquartile range [q25, q75] shown in brackets. * N.A. = Not Applicable.

**Table 2 cancers-17-01251-t002:** Confusion matrix of Training dataset.

		Actual		Total
		OC	Benign	
Predict	OC	12	0	12
	Non-OC	1	16	17
Total		13	16	29

**Table 3 cancers-17-01251-t003:** Confusion matrix of Test dataset.

		Actual		Total
		OC	Benign	
Predict	OC	4	3	7
	Non-OC	0	18	18
Total		4	21	25

**Table 4 cancers-17-01251-t004:** Confusion matrix of total samples.

		Actual			Total
		OC	Benign	AC	
Predict	OC	16	3	1	20
	Non-OC	1	34	33	68
Total		17	37	34	88

## Data Availability

The datasets analyzed for the current study are available from the corresponding author upon reasonable request to the corresponding author, TaeJin Ahn.

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
