# Peer review of "Innovative qPCR Algorithm Using Platelet-Derived RNA for High-Specificity and Cost-Effective Ovarian Cancer Detection"

_cancers, 2025, doi:10.3390/cancers17071251_

Round 1

Reviewer 1 Report

Comments and Suggestions for Authors

Ahn et al., exploited specific splice variants in platelet RNA as biomarkers and developed a qPCR-based algorithm to detect these markers. In contrast to expensive NGS technology, qPCR offers a more cost-effective solution while demonstrating exceptional specificity through its advanced algorithm.. It can effectively distinguish between ovarian cancer and non-ovarian cancer cases, thereby improving the specificity and cost-effectiveness of ovarian cancer detection. The results showed that identified RNA markers that can distinguish ovarian cancer from benign diseases with more than 94% accuracy. This technology can improve early detection rates and progress to liquid slide diagnostics. The novelty and contribution of this paper included:

  1. Applied cell-specific splice junctions of platelet RNA as biomarkers of malignant ovarian tumor.
  2. Identified 10 markers suitable for detection of ovarian cancer
  3. Clearly demonstrates that their qPCR method is cost-effective compared to technologies such as next-generation sequencing (NGS), making it easier to implement broadly in the clinic

Generally, this report is well-conducted, and the results support the title. Due to the accuracy and cost-effective characteristics of the biomarkers and qPCR-based algorithm, it might arouse readers' interest.

However, too many abbreviations do not explain. Please spell out the full name of all non-standard abbreviations on first use.  

Author Response

Comment 1:  Too many abbreviations do not explain. Please spell out the full name of all non-standard abbreviations on first use.  

Response 1: Thank you for your thoughtful comment. We understand that our marker names might be somewhat confusing, and we agree that the naming conventions should be clearer for readers. To address this, we have added the following sentence to the results section (lines 301–302). "We noted a list of markers that follow the gene name with the suffix “_a” or “_b.”

Reviewer 2 Report

Comments and Suggestions for Authors

The manuscript entitled “Innovative qPCR Algorithm Using Platelet-Derived RNA for High-Specificity and Cost-Effective Ovarian Cancer Detection” presents an approach for early ovarian cancer detection using a qPCR-based algorithm applied to platelet-derived RNA. The proposed method demonstrates high diagnostic accuracy (AUC = 0.933). The strategy is novel and clinically relevant; however, several methodological concerns limit the interpretability and reproducibility of the findings.

The Introduction section of the manuscript is well-structured and provides a solid foundation for the study; however, it does not mention existing TEP-based diagnostic models in other cancers, which might be a good introduction to the problem.

The Materials and Methods section is appropriate for the study’s objectives. The Authors demonstrate careful experimental planning, and the methodologies employed are consistent with current best practices in the field of RNA biomarker discovery. Overall, the methods are well-executed and suitable for the study's aims.

The Results section is clearly structured, and the authors have presented their findings logically and convincingly. The combination of RNA sequencing, qPCR validation, and algorithm-based classification is well-executed, and the data presented support the conclusions.

The Discussion section generally supports the results and discusses them in relation to other works. Firstly, the Discussion highlights and properly discusses how the qPCR-based algorithm achieves high sensitivity and specificity and connects several markers (e.g., IL1R2, DEFA1, CD177) to known immune or cancer-related functions. However, the authors should be cautious with statements that this method “outperforms conventional CA125-based screening.” The Authors did not conduct a direct comparison, which is necessary for substantiating this claim.

Concerns and Suggestions:

  1. The number of ovarian cancer patients, particularly in the test set (n = 4), is small.
  2. Significant age differences between the control and cancer groups are not adjusted for, which may bias classification.
  3. The test set lacks complete independence; controls were involved in marker discovery.
  4. As samples were collected from multiple institutions, it is important to clarify whether batch effects were assessed or corrected in the analysis.
  5. Primer sequences, qPCR efficiencies, and platelet purity metrics are missing, limiting reproducibility.
  6. The role of the two borderline tumour patients is a little bit unclear. Although they were included in the validation, it would be helpful to comment on how their inclusion may influence the interpretation of classification performance.

It should, however, be underlined that the presented study has strong potential and offers meaningful advances in non-invasive ovarian cancer diagnostics.

Author Response

Comment 1: The number of ovarian cancer patients, particularly in the test set (n = 4), is small.

Response 1: We agree with the reviewer’s comment regarding the small number of ovarian cancer patients in the test set. This limitation has been explicitly acknowledged in the revised manuscript at lines 410-414.

Comment 2: Significant age differences between the control and cancer groups are not adjusted for, which may bias classification.

Response 2: We agree that the age difference between groups may influence classification and appreciate the reviewer highlighting this limitation. In response, we have updated the Discussion section to address this point. It has been addressed in the revised manuscript at lines 417-420.

Comment 3: The test set lacks complete independence; controls were involved in marker discovery.

Response 3: We agree that the reuse of control samples in both marker discovery and testing may affect the independence of the test set. This limitation has been incorporated in the revised manuscript at lines 421–424

Comments 4: As samples were collected from multiple institutions, it is important to clarify whether batch effects were assessed or corrected in the analysis.

Response 4: In response to the reviewer's insightful comment, we analyzed the batch effect across multiple institutions and found no statistically significant difference in the ACTB signal among them. Therefore, different collection sites can likely be merged without adjusting for batch effects. The analysis results are now presented in the Results section (lines 275–290) and Figure 1.

Comments 5: Primer sequences, qPCR efficiencies, and platelet purity metrics are missing, limiting reproducibility.

Due to a conflict of interest and an ongoing patent application, we have not provided the primer sequences. However, to ensure transparency regarding qPCR efficiencies and platelet purity, we have included the Cq values for ACTB PCR and the RNA integrity numbers for all samples. Please refer to Supplementary Table 1 for details.

Comment 6: The role of the two borderline tumour patients is a little bit unclear. Although they were included in the validation, it would be helpful to comment on how their inclusion may influence the interpretation of classification performance.

Response 6: We agree that clarification regarding the inclusion of borderline ovarian tumor patients is necessary and appreciate the reviewer’s attention to this point. With regard to the inclusion of borderline ovarian tumors, we have added the following clarification to the revised manuscript in the Discussion section at lines 424-427.